# *Cis*-regulatory variants affect gene expression dynamics in yeast

**Ching-Hua Shih, Justin Fay\***

Department of Biology, University of Rochester, Rochester, United States

**Abstract** Evolution of *cis*-regulatory sequences depends on how they affect gene expression and motivates both the identification and prediction of *cis*-regulatory variants responsible for expression differences within and between species. While much progress has been made in relating *cis*-regulatory variants to expression levels, the timing of gene activation and repression may also be important to the evolution of *cis*-regulatory sequences. We investigated allele-specific expression (ASE) dynamics within and between *Saccharomyces* species during the diauxic shift and found appreciable *cis*-acting variation in gene expression dynamics. Within-species ASE is associated with intergenic variants, and ASE dynamics are more strongly associated with insertions and deletions than ASE levels. To refine these associations, we used a high-throughput reporter assay to test promoter regions and individual variants. Within the subset of regions that recapitulated endogenous expression, we identified and characterized *cis*-regulatory variants that affect expression dynamics. Between species, chimeric promoter regions generate novel patterns and indicate constraints on the evolution of gene expression dynamics. We conclude that changes in *cis*-regulatory sequences can tune gene expression dynamics and that the interplay between expression dynamics and other aspects of expression is relevant to the evolution of *cis*-regulatory sequences.

**\*For correspondence:**
justin.fay@rochester.edu

**Competing interests:** The authors declare that no competing interests exist.

## Introduction

Noncoding *cis*-regulatory sequences control gene expression and are thought to play a central role in evolution (*Carroll, 2005*). However, genetic analysis of phenotypic variation frequently uncovers changes in protein coding sequences rather than in regulatory regions (*Stern and Orgogozo, 2008*; *Fay, 2013*). One explanation might be that genetic mapping and transgenic studies tend to identify large effect mutations. If evolution predominantly occurs through numerous changes of small effect (*Rockman, 2012*), the role of *cis*-regulatory sequences is harder to discern. Even so, *cis*-regulatory changes have been shown to be important in polygenic adaptation (*Bullard et al., 2010*; *Fraser et al., 2010*; *Fraser et al., 2011*; *Fraser et al., 2012*; *Naranjo et al., 2015*), the accumulation of multiple changes at evolutionary hotspots (*Frankel et al., 2011*; *Engle and Fay, 2012*, 1; *Martin and Orgogozo, 2013*; *Li and Fay, 2019*), and variation in fitness and disease (*Boyle et al., 2017*; *Sharon et al., 2018*). Regardless of the relative role of coding and noncoding sequences in phenotypic evolution, understanding how variation in *cis*-regulatory sequences generates variation in gene expression is important to understanding the evolution of gene regulation.

Across organisms, there is an abundance of *cis*-acting sequence variation that affects gene expression levels (*Hill et al., 2021*). The causes of *cis*-regulatory variation are not as easily characterized. Transcription factor binding sites are often found to play important roles (*Zheng et al., 2011*). However, the number, identity, and position of binding sites can also vary without affecting expression. The flexibility of *cis*-regulatory sequences is shown by genes with similar expression patterns but different *cis*-regulatory sequences (*Berman et al., 2002*). In the case of orthologous genes from different species, binding site turnover and transcription factor re-wiring explain substantial divergence in *cis*-regulatory sequences without expression divergence (*Ludwig et al., 2000*;

*Dermitzakis and Clark, 2002*; *Hare et al., 2008*; *Tuch et al., 2008*; *Venkataram and Fay, 2010*; *Swanson et al., 2011*; *Bergen et al., 2016*). Consequently, predicting changes in gene expression based on variation in individual transcription factor binding sites has proven difficult (*Doniger and Fay, 2007*; *Doniger et al., 2008*).

Despite the flexibility of binding sites within *cis*-regulatory sequences, sequences flanking binding sites evolve under constraints and can affect expression. For example, over a third of yeast intergenic sequences are estimated to be under selective constraint (*Chin et al., 2005*; *Doniger et al., 2005*). This fraction is greater than that expected from either conserved or experimentally identified binding sites (*Doniger et al., 2005*; *Venkataram and Fay, 2010*). Sequences flanking binding sites have also been shown to affect expression, potentially related to DNA shape, nucleosome positioning, or weak binding sites (*Tanay et al., 2005*; *White et al., 2013*; *Abe et al., 2015*; *Levo et al., 2015*; *Inukai et al., 2017*). Consequently, conservation scores have proven important for predicting *cis*-regulatory variants (*Huang et al., 2017*; *Kircher et al., 2019*; *Renganaath et al., 2020*).

*Cis*-regulatory sequences can affect other aspects of gene regulation besides expression levels. Stochastic noise in gene expression provides a mechanism for bet-hedging strategies (*Raj and van Oudenaarden, 2008*) and is encoded by and evolves through changes in *cis*-regulatory sequences (*Richard and Yvert, 2014*). *Cis*-regulatory variants that alter noise in expression levels have been shown to be under selection and can occur both within and outside of known binding sites (*Carey et al., 2013*; *Sharon et al., 2014*; *Metzger et al., 2015*; *Schor et al., 2017*; *Duveau et al., 2018*).

Gene expression dynamics, which include the timing and rate of gene activation and repression, are also important aspects of gene regulation (*López-Maury et al., 2008*; *Yosef and Regev, 2011*). Gene expression dynamics can be altered by transcription factors and their interactions with promoters, but also depend on nucleosomes and their positions relative to binding sites (*Lam et al., 2008*; *Hager et al., 2009*; *Dadiani et al., 2013*; *Hansen and O'Shea, 2015*). Notably, chromatin mutants slow gene activation without compromising final levels of gene expression (*Barbaric et al., 2001*; *Floer et al., 2010*). Variation in *cis*-regulatory sequences can also affect gene expression dynamics, but these dynamics are only sometimes captured (*Ackermann et al., 2013*; *Francesconi and Lehner, 2014*; *Strober et al., 2019*). Thus, the causes of *cis*-regulatory variation in gene expression dynamics have neither been characterized nor related to variation in gene expression levels.

In this study, we investigate *cis*-acting variation in gene expression dynamics. We survey and find allele-specific differences in expression dynamics both within and between *Saccharomyces* species during the diauxic shift when there is major transition from the expression of genes involved in fermentation to respiration (*DeRisi et al., 1997*). Using these data, we associated allele-specific expression (ASE) with promoter variation and individual variants using a high-throughput reporter assay. Our results inform our understanding of variation in gene expression dynamics and point towards an integrated view of gene expression and how it evolves.

## Results

### *Cis*-regulatory variation in gene expression levels and dynamics

To identify *cis*-regulatory variation in gene expression dynamics, we measured ASE in three intraspecific and two inter-specific diploid hybrids. Hybrids were generated by crossing a North American *Saccharomyces cerevisiae* strain (Oak) to an *S. cerevisiae* wine strain (Wine) and two strains from China (China I and China II), as well as to a strain of *Saccharomyces paradoxus* and *Saccharomyces uvarum* (Table S1 in *Supplementary file 1*), enabling us to examine a range of divergence in gene regulation. To capture temporal differences in ASE that occur during the diauxic shift, we generated RNA-sequencing data from 19 timepoints for each hybrid, spanning the shift from fermentation to respiration as measured by glucose depletion (*Figure 1—figure supplement 1*).

ASE requires RNA-sequencing reads that can be distinguished as coming from one of the two parental strains. To measure ASE while avoiding mapping bias (*Degner et al., 2009*; *Stevenson et al., 2013*), we mapped reads to the combined parental genomes and enumerated allele-specific reads. The proportion of reads mapping to each parental genome was equivalent

across all five hybrids, except for one arm of chromosome XIII in the China I hybrid consistent with aneuploidy (*Figure 1—figure supplement 2*), which we removed from subsequent analysis.

Two statistical tests were used to separately identify genes exhibiting differences in ASE levels and changes in ASE dynamics over time. Genes with ASE dynamics were identified by testing for an autocorrelation in the ratio of allele-specific reads over time. Under the null model, the ratio of the two alleles is constant over time but not necessarily equal to 1. Genes with ASE levels were identified by testing for differences between the expression of the two alleles across all timepoints. Using these tests, we found that more genes showed ASE levels compared to ASE dynamics and the number of genes with either ASE levels or dynamics increased with divergence (false discovery rate [FDR] < 0.01, *Table 1*). As expected, genes with ASE levels showed larger average allele differences across timepoints, and genes with ASE dynamics showed larger standard deviations in allele differences across timepoints (*Figure 1—figure supplement 3*). Genes with ASE levels and genes with ASE dynamics were relatively evenly distributed across genes whose expression increased/decreased or showed a peak/trough during the diauxic shift (*Figure 1* in *Supplementary file 1*).

Changes in ASE over time can result from a variety of differences in the dynamics of the two alleles. To illustrate this variety, we consider a gene that is activated during the diauxic shift (*Figure 2A, B*). ASE can be condition-specific due to an allele difference in the presence but not absence of glucose, or vice versa. ASE can also differ specifically during the diauxic shift due to a difference in the timing or rate of gene activation that does not require ASE differences before or after the shift. To characterize ASE, we applied k-means clustering to ASE allele frequencies and found two types of patterns (*Figure 2C*, Table S3 in *Supplementary file 1*). The majority of genes (72%) showed environment-dependent ASE and the remaining genes showed an ASE maximum or minimum during the transition (clusters 6, 9, 10, and 12, *Figure 2C*).

## ASE is associated with SNPs and InDels

ASE is caused by *cis*-acting single-nucleotide polymorphisms (SNPs) or insertion deletion polymorphisms (InDels) that affect gene expression. In yeast, *cis*-acting variants most likely occur within the small (~500 bp) intergenic region upstream of a gene, but could also occur within the coding or 3′ region of a gene. For each hybrid, we tested whether the number of variants in these regions predicts significant ASE levels or ASE dynamics using logistic regression.

Both ASE levels and dynamics were associated with the number of SNP and InDel variants, but these associations varied by hybrid and the type of variant. Within intra-specific hybrids, upstream SNPs and InDels were associated with both ASE levels and dynamics, but InDels showed stronger associations as measured by the odds ratio and the significance (*Figure 3*). SNPs and InDels within coding and downstream regions were also associated with ASE, but the associations were similar for ASE levels and dynamics (Table S4 in *Supplementary file 1*). Although the odds ratio is high for downstream variants, this could be due to either larger effects or a higher fraction of functional variants within the 80 bp of downstream sequence examined. To examine the magnitude of ASE differences, we split the genes into three groups and found that upstream InDels have a similar ability to predict ASE levels and dynamics for genes with large ASE differences, but predict ASE dynamics better than ASE levels for genes with small ASE differences (*Figure 3—figure supplement 1*).

**Table 1.** Number of genes with allele-specific expression.

| | Intra-specific hybrids[‡] | | | Inter-specific hybrids[‡] | |
|---|---|---|---|---|---|
| | *S. cerevisiae* (Oak × Wine) | *S. cerevisiae* (Oak × China II) | *S. cerevisiae* (Oak × China I)† | *S. cerevisiae* × *S. paradoxus* | *S. cerevisiae* × *S. uvarum* |
| Group[*] | YJF1460 | YJF1455 | YJF14542 | YJF1453 | YJF1484 |
| Dynamics | 671 | 699 | 911 | 2055 | 1827 |
| Levels | 1964 | 2088 | 2260 | 2930 | 3237 |
| Both | 371 | 375 | 501 | 1260 | 1253 |

[*]Genes with significant (false discovery rate < 0.01) allele-specific differences in dynamics, levels, or both dynamics and levels.

[†]The total number of genes is 4703 except for 358 genes on chromosome 13R of the China I hybrid that were removed.

[‡]Oak is most closely related to the Wine strain, followed by China II, China I, *S. paradoxus,* and *S. uvarum.*

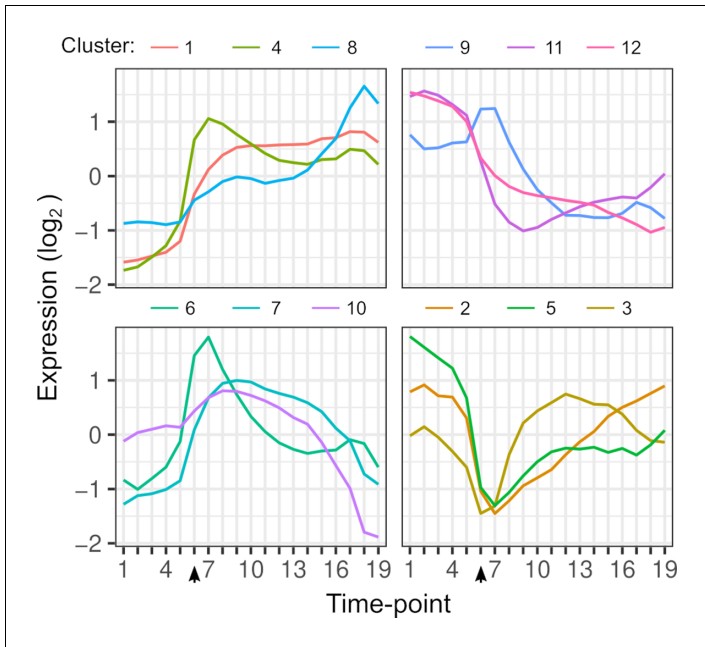

**Figure 1.** Gene expression dynamics. Each line shows the average expression of genes in each k-means cluster over timepoints. Clustering is based on the expression of 4703 genes from each hybrid. The arrow indicates the timepoint when glucose was depleted.

The online version of this article includes the following figure supplement(s) for figure 1:

**Figure supplement 1.** Sampling scheme for gene expression dynamics during the diauxic shift.
**Figure supplement 2.** Chromosome 13R aneuploidy in YJM1454 (Oak × ChI).
**Figure supplement 3.** Characteristics of differentially expressed genes.

Inter-specific hybrids only showed significant associations between ASE dynamics and upstream SNPs and InDels. Association between ASE and divergence may be weak or absent if most substitutions between species do not affect gene expression or if there are many substitutions that affect expression but they have random effects that cancel each other out. The inter-specific hybrids have much higher rates of divergence compared to the intra-specific hybrids: an average of 15 and 107 upstream InDels and SNPs, respectively, compared to 1.2 and 6.0 InDels and SNPs within the intra-specific hybrids (Table S5 in *Supplementary file 1*). For the intra-specific hybrids, the frequency of ASE increases linearly with the number of variants (*Figure 3—figure supplement 2*), suggesting that regulatory variants are rare enough to consistently increase the frequency of ASE at low divergence.

### Intra-specific *cis*-regulatory variants

The associations between ASE and the number of upstream intergenic variants indicate that promoter polymorphism is a significant contributor to ASE. To specifically measure the effects of promoter polymorphism and identify causal variants, we used a high-throughput <u>cis</u>-regulatory <u>e</u>lement (CRE-seq) reporter assay (*Mogno et al., 2013*). In this assay, promoter sequences are synthesized and the resulting pooled library is cloned and integrated into a single site in the yeast genome (*Figure 4—figure supplement 1*). The synthesized sequences include a 10 bp barcode that can be used as a tag to measure gene expression through RNA-sequencing and relative abundance through DNA-sequencing.

We designed a CRE-seq library to test promoter variants upstream of 69 genes that exhibited ASE levels and/or dynamics in the Oak × ChII hybrid. Because the synthesized promoters were limited to 130 bp, we designed five overlapping CRE sequences per gene to test all variants within the 250 bp region upstream of the transcription start site (TSS). There were a total of 337 variants, an average of 4.2 SNPs and 0.72 InDels per gene. For any CRE sequences with more than a single difference between the Oak and ChII alleles, we also generated CREs for each ChII variant in the Oak allele and vice versa (*Figure 4—figure supplement 1*). The total library contained 1818 CREs with

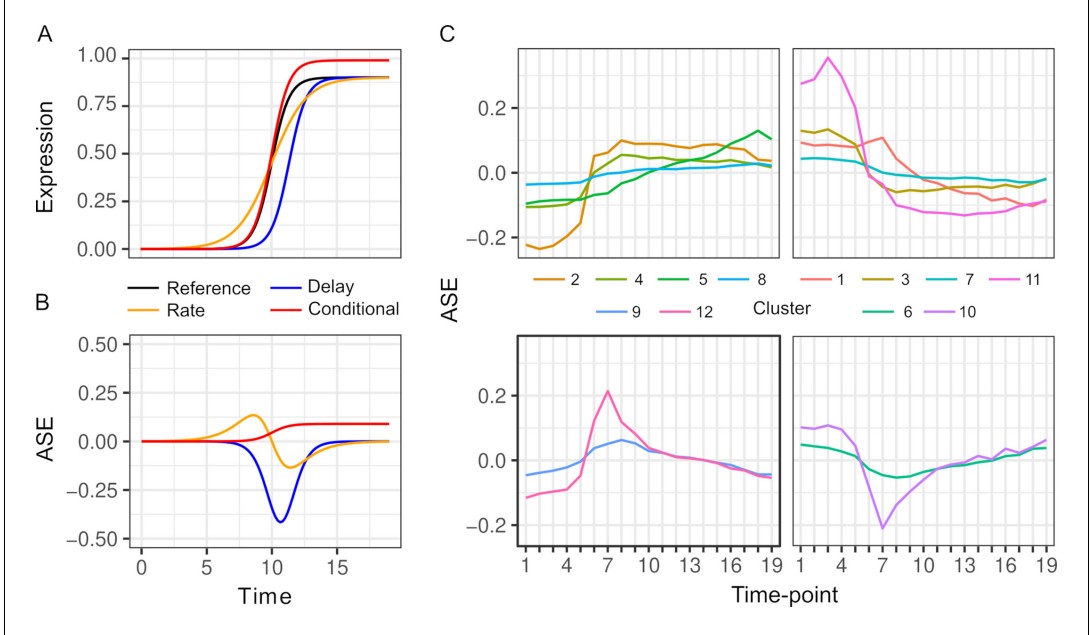

**Figure 2.** Patterns of allele-specific expression (ASE) dynamics. (**A**) Three hypothetical types of differences in expression dynamics in comparison to a common reference (black) are shown by a time delay (blue), rate change (orange), and condition-specific expression difference (red). (**B**) ASE based on the three types of differences in comparison to the common reference. (**C**) Average ASE across 19 timepoints of k-means clustering of 6135 genes with significant ASE dynamics. Clusters 6, 9, 10, and 12 (bottom panels) show maximum deviation during the diauxic shift, whereas the others generally show increasing or decreasing ASE differences over time consistent with condition-specific ASE.

four barcode replicates per CRE and included all variants within the 334 regions upstream of the 69 genes.

We first tested whether the synthetic promoter regions could recapitulate expression of the endogenous genes. We measured CRE expression over 19 timepoints during the diauxic shift in the same hybrid background as RNA-seq. Out of 334 regions, 137 were correlated with RNA-seq expression (FDR < 0.05) and 17 genes had no region correlated. CRE-seq regions correlated with RNA-seq tended to lie further away from the TSS, and in some cases showed a gradual increase in correspondence (*Figure 4*). We also barcoded and measured expression from full-length promoters of three genes (*Figure 4—figure supplement 2*). All three genes showed good correspondence between the full-length promoter and a shorter CRE-seq region. However, one of the genes, *ALD5*, showed both short- and full-length reporter expression notably different from the endogenous RNA-seq expression. One explanation for this difference is that we used an annotated TSS 515 bp upstream of the ATG, whereas some studies indicate a site much closer (78–84 bp) to the ATG (*Zhang and Dietrich, 2005*; *Pelechano et al., 2013*).

To determine whether any of the CRE regions contained variants that affect expression, we examined the 281/334 regions upstream of the 69 genes with one or more differences between the Oak and ChII alleles. One of the regions showed significant differences in expression levels between the Oak and ChII alleles, and 31 showed differences in expression dynamics (FDR < 0.05, *Table 2*). Eleven of these regions had only a single variant that differentiated the two parental alleles and the rest had between 2 and 9 variants. For regions with multiple variants, we tested each using CREs containing the Oak variant in the ChII background and vice versa. For expression levels, we found significant effects for one of the two variants tested, and for expression dynamics we found 35 out of 70 variants (FDR < 0.05, *Table 2*), 4 of which were detected by multiple regions.

Promoter regions that showed allele-specific differences in expression dynamics had high rates of polymorphism and often multiple variants that affected expression. The rate of variants in the 31 CRE regions with differences in expression dynamics (2.1%) was higher than that of intergenic regions across the genome (1.4%; Fisher's exact test, p=0.0038). Out of the 22 genes with one or more CRE regions showing differences in expression dynamics between the Oak and ChII alleles, 8

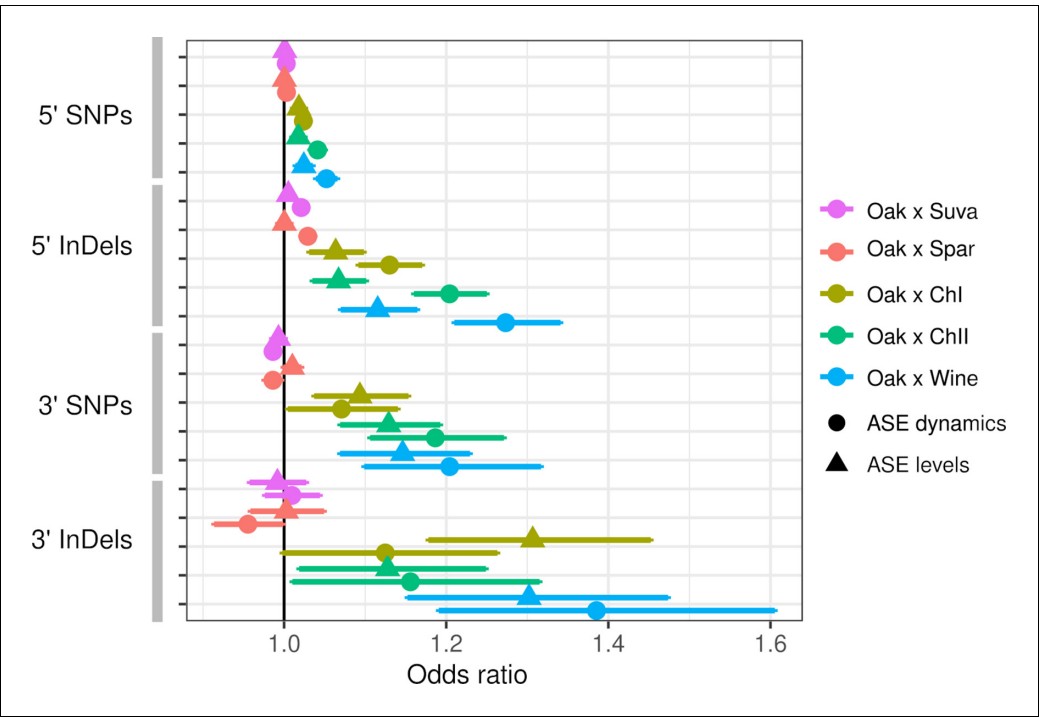

**Figure 3.** Allele-specific expression (ASE) is associated with intergenic single-nucleotide polymorphisms (SNPs) and insertions/deletions (InDels). The odds ratio (OR) and 95% confidence interval for associations between the number of SNPs or InDels and significant ASE levels (triangles) and dynamics (circles). The OR of each hybrid is shown separately for upstream (5') and downstream (3') intergenic variants.

The online version of this article includes the following figure supplement(s) for figure 3:

**Figure supplement 1.** Allele-specific expression (ASE) associations with single-nucleotide polymorphisms (SNPs) and insertions/deletions (InDels) for genes with small (**A**), medium (**B**), and large (**C**) expression differences.

**Figure supplement 2.** The frequency of significant allele-specific expression (ASE) dynamics (**A**) and ASE levels (**B**) as a function of the number of variants.

had two or more significant variants, 12 had only a single variant, and 2 had no significant variants. While the number of InDels with significant effects on expression dynamics was small, the ratio of significant SNPs to InDels (6.0) was not different from that of intergenic regions across the genome (4.8; Fisher's exact test, p>0.05).

Variants with significant effects on CRE-seq expression were not always consistent with patterns of RNA-seq ASE (*Figure 5*). Only 16 of the 31 regions showing CRE-seq expression dynamics correlated with RNA-seq expression. For example, region 4 of the *YPS6* promoter showed increased expression over time in the RNA-seq data but not in the CRE-seq data. Even so, the Oak allele exhibited higher expression levels than the ChII allele in both the RNA-seq and CRE-seq assays, a difference that can be attributed to one of the two variants in the region. The *ICL2* promoter showed increased expression over time in both the RNA-seq and CRE-seq assays, but not all CRE-seq allele differences were consistent with the RNA-seq allele differences. Region 1 of the *ICL2* promoter showed ASE differences consistent with RNA-seq and multiple variants with effects on expression dynamics. However, region 3, which had only a single variant, showed allele differences in expression dynamics inconsistent with RNA-seq, whereby the Oak allele responded more strongly than the ChII allele to the diauxic shift.

*Cis*-regulatory variants that affect expression levels have been associated with conserved promoter regions and disruption of transcription factor binding sites (*Renganaath et al., 2020*). We found no difference in PhastCons conservation scores or change in binding site scores between the 35 variants associated with expression dynamics and those that were not associated (*Figure 5—figure supplement 1*). Because the number of variants is small, we also examined whether PhastCons scores or binding site scores improved the genome-wide logistic regression. Binding site scores did

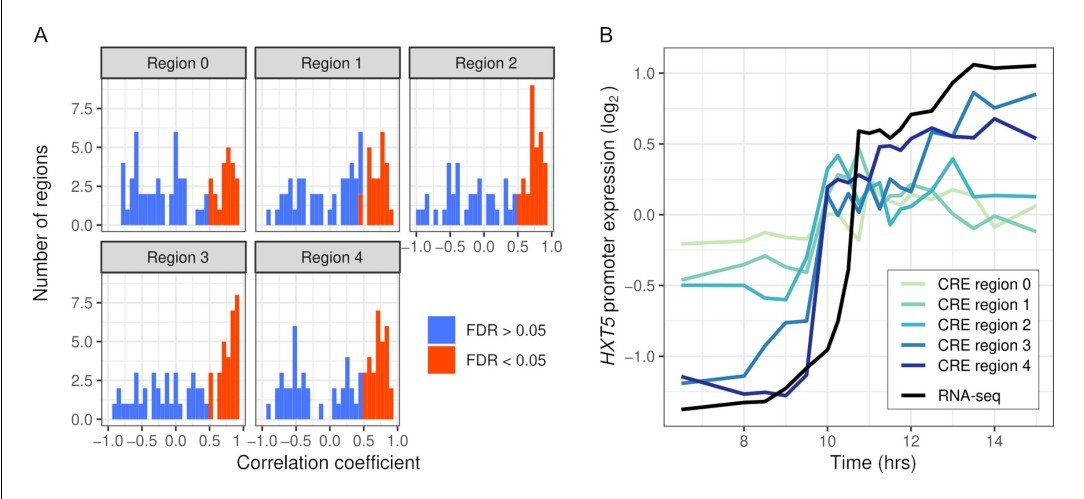

**Figure 4.** Intra-specific *cis*-regulatory elements (CREs) recapitulate endogenous expression dynamics. (**A**) Histogram of the correlations between CRE expression and the endogenous (RNA-seq) expression of 69 genes. Correlations are shown separately for CRE regions 0–4, ordered proximal to distal of the transcription start site with shifts of 30 bp. CREs with a significant (false discovery rate < 0.05) correlation are shown in red, the rest in blue. (**B**) CRE expression of five regions upstream of *HXT5* as well as its endogenous expression from RNA-seq.

The online version of this article includes the following figure supplement(s) for figure 4:

**Figure supplement 1.** Design and cloning of *cis*-regulatory element (CRE-seq) libraries.

**Figure supplement 2.** Short *cis*-regulatory elements (CREs) recapitulate longer CRE expression.

not improve the odds ratios. While PhastCons scores improved the association between SNPs and ASE levels and dynamics, it did not improve InDel associations (Table S6 in *Supplementary file 1*).

## Inter-specific *cis*-regulatory variation

We also designed a CRE-seq library to test promoter divergence of 98 genes that exhibited ASE levels and/or dynamics in the *S. cerevisiae* × *S. uvarum* hybrid. We used the same design of five overlapping 130 bp CRE sequences covering 250 bp upstream of the TSS. There was an average of 32 substitutions and 4.0 InDels per region. Because there were too many differences to test individually, we generated chimeric CRE sequences containing either the first or second half of the sequence from the *S. cerevisiae* allele and the remaining half from the *S. uvarum* allele (*Figure 4—figure supplement 1*). The total library contained 1808 CREs with four barcode replicates per CRE and covered 452 regions upstream of the 98 genes.

We again tested whether the synthetic promoter regions could recapitulate expression of the endogenous genes and whether there were differences in expression levels or dynamics between the *S. cerevisiae* and *S. uvarum* alleles. We measure CRE expression during the diauxic shift in the same hybrid background used to measure RNA-seq. Out of 452 regions, 220 were correlated with RNA-seq expression (FDR < 0.05) and 28 genes had no region correlated. Similar to intra-specific

**Table 2.** CRE regions and variants affecting gene expression.

| Library | Type | Genes | Regions | SNPs | InDels |
|---|---|---|---|---|---|
| Intra-specific | Levels | 1/59 | 1/240 | 0/1 | 1/1 |
| Intra-specific | Dynamics | 22/50 | 31/201 | 30/57 | 5/13 |
| Inter-specific | Levels | 2/86 | 2/317 | -/68 | -/12 |
| Inter-specific | Dynamics | 59/72 | 113/257 | -/3560 | -/479 |

Genes, regions, SNPs, and InDels are the number significant out of the number tested. Individual SNPs and InDels were not tested for the inter-specific library.

CRE: *cis*-regulatory element; SNPs: single-nucleotide polymorphisms; InDels: insertions/deletions.

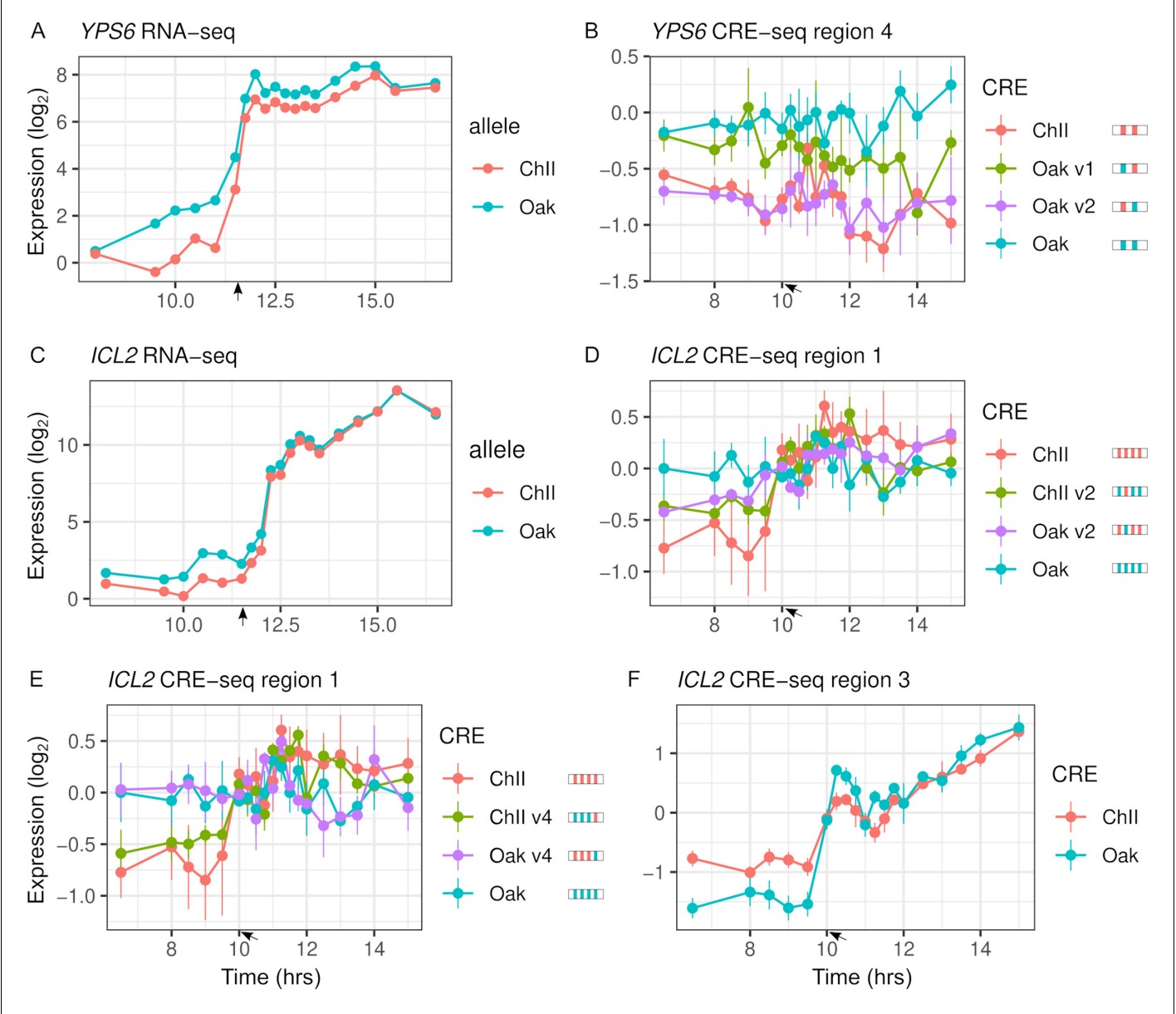

**Figure 5.** Intra-specific *cis*-regulatory elements (CREs) show differences in expression levels and dynamics. (**A**) *YPS6* shows allele differences in endogenous expression levels and dynamics. (**B**) CRE region 4 of the *YPS6* promoter shows allele differences in expression levels, but is not correlated with endogenous expression patterns. Substituting the Oak insertions/deletions (InDel) into the ChII allele (Oak v1) increases expression levels, but substituting the Oak single-nucleotide polymorphism (SNP) into the ChII allele (Oak v2) has no effect. (**C**) *ICL2* shows allele differences in endogenous expression dynamics. Of the four SNPs and one InDel that differentiate the region 1 CRE alleles, two SNPs (v2 and v4) alter expression dynamics in both the Oak and ChII background (**D, E**). (**F**) CRE region 3 of *ICL2* has a single InDel between the Oak and ChII alleles and also shows allele differences in expression dynamics. For panels (**B**), (**D**), and (**E**), CRE alleles are shown by rectangles with colored ticks to indicate the Oak and ChII variants. Bars indicate standard errors. Arrows indicate the approximate time of glucose depletion.

The online version of this article includes the following figure supplement(s) for figure 5:

**Figure supplement 1.** Binding site and conservation scores of variants.

comparisons, more regions showed differences in expression dynamics (113) than expression levels (2) between the two species' alleles (*Table 2*).

Expression driven by chimeric sequences may lie within the range of the two parental species and can be used to map parental differences to the proximal or distal portion of the *cis*-regulatory region (*Figure 6—figure supplement 1*). However, chimera expression may also lie outside of the parental

range if recombination brings together variants with effects in the same direction or if there are epistatic interactions between variants. Such *cis*-regulatory interactions are thought to be common due to binding site turnover (*Zheng et al., 2011*) and do not require expression divergence between the parental species.

To map expression divergence and identify chimeras outside of the parental range, we tested each of the two chimeras for differences with each parent. Out of the 113 regions with parental species differences in expression dynamics, 57 were consistent with the proximal and 14 were consistent with the distal region explaining the difference. For example, the proximal region of *SDH4* can explain the entire difference between the parental species' alleles (*Figure 6A, D*). Examining all the regions (n = 348), 88 chimeras showed expression dynamics that differed from both parents (FDR < 0.05). The majority of these chimeras are not intermediate between the two parents; in 63 cases, the expression distance between the two parents was less than the average distance of the chimera to either parent, and in 56 cases there was no difference between the two parents. As examples, *MDM36* shows chimera expression between the two parents (*Figure 6B, E*), and *IDP2* shows chimera expression outside the parental range (*Figure 6C, F*). For the 63 chimeras with high expression

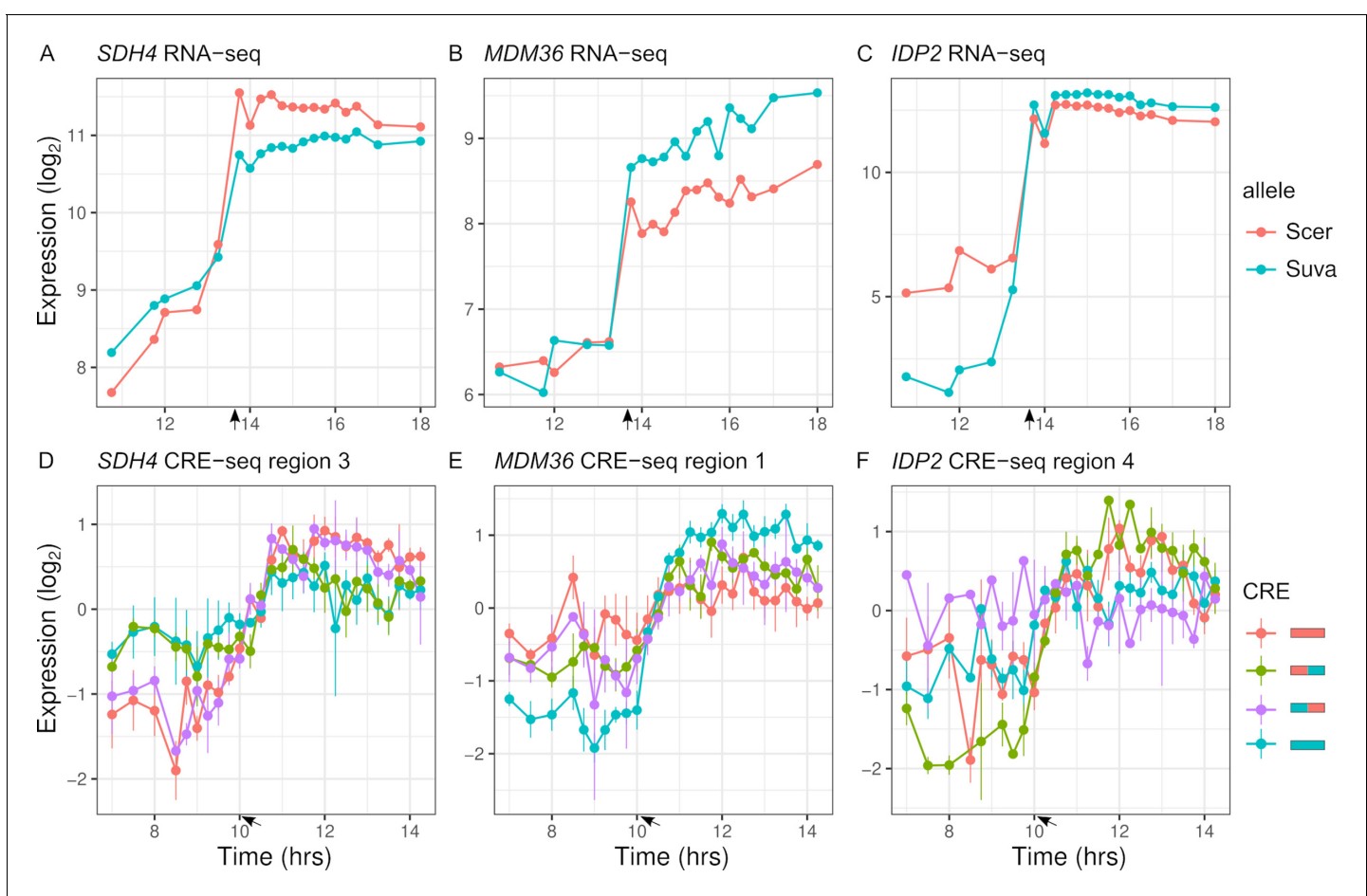

**Figure 6.** Inter-specific *cis*-regulatory elements (CREs) show differences in expression dynamics. (**A–C**) Endogenous expression of *S. cerevisiae* (Scer) and *S. uvarum* (Suva) alleles of *SDH4, MDM36,* and *IDP2*. (**D–F**) CRE-seq expression of region 3 (*SDH4*), region 1 (*MDM36*), and region 4 (*IDP2*) for parental *S. cerevisiae* (red) and *S. uvarum* (blue) CRE alleles and both chimeric CREs. *SDH4* shows expression divergence maps to the proximal promoter region, *MDM36* and *IDP2* show chimera expression that differ from both parents, with the *MDM36* chimeras being between the two parents and *IDP2* chimeras being outside the two parents. Bars indicate standard errors. Arrows indicate the approximate time of glucose depletion. The online version of this article includes the following figure supplement(s) for figure 6:

**Figure supplement 1.** Identification of significant differences for the intra-specific and inter-specific *cis*-regulatory element sequence (CRE-seq) libraries.

distance from both parents, the expression of the chimera was outside the two parental values for most (21.6/27) of the timepoints.

## Discussion

*Cis*-regulatory sequences control the activation and repression of genes in response to cellular and environmental signals, and the consequences of variation within these sequences are relevant to our understanding of evolution and human disease. Much progress has been made in identifying *cis*-regulatory variants responsible for changes in gene expression levels. In this study, we identified and characterized *cis*-acting variation in gene expression dynamics. We find that while gene expression dynamics are interrelated to expression levels, they differ in the types and identities of variants perturbing them. Below we discuss the relationship between gene expression dynamics and other aspects of gene regulation and how this fits into our understanding of regulatory evolution.

### Gene expression dynamics are an integrated component of *cis*-regulatory variation

We find, as expected, abundant *cis*-acting variation in gene expression dynamics. Despite the potential importance of changes in gene expression dynamics, they are not easily disentangled from other aspects of gene expression such as levels and noise. In our data, roughly one-third of genes that exhibit ASE dynamics also exhibit ASE levels (*Table 1*). This overlap is not unexpected as the tests for ASE dynamics and levels are not mutually exclusive. For example, genes that exhibit differences in expression levels after but not before the diauxic shift by definition also exhibit differences in expression dynamics during the diauxic shift. While less common, we do find genes where ASE differences are greatest during the diauxic shift, consistent with a change in the rate or timing of gene activation/repression (*Figure 2*). How can the rate or timing of expression be altered without affecting levels? Prior work has shown that chromatin mutants can slow gene activation without impacting levels (*Barbaric et al., 2001*; *Floer et al., 2010*). While this does not exclude the possibility of changes in binding sites, it points to changes in nucleosome positioning sequences in mediating expression dynamics.

Gene expression dynamics may also be interrelated to stochastic noise in expression. During the diauxic shift, there is cell-to-cell heterogeneity in gene expression and this leaky regulation can give the appearance of a slower rate of activation at the population level (*New et al., 2014*; *Venturelli et al., 2015*). Put differently, an increase in noise is expected to blur a sharp transition between activated and repressed states. However, noise, levels, and dynamics are also not entirely dependent on one another. For example, gene expression often occurs in bursts with expression levels more dependent on burst size, noise more dependent on burst frequency (*Cai et al., 2006*; *Carey et al., 2013*), and timing dependent on both changes in burst size and frequency. Thus, the covariation between expression levels, dynamics, and noise likely shapes how gene expression evolves within and between species.

### *Cis*-regulatory variants underlying expression dynamics

Previous studies have characterized *cis*-regulatory variants underlying gene expression levels. We find a number of results from studies of gene expression levels also hold for expression dynamics: (i) a weak or no association between expression divergence and sequence divergence between species (*Tirosh et al., 2008*; *Zeevi et al., 2014*; *Li and Fay, 2017*); (ii) 5′ InDels are more strongly associated with expression differences than 5′ SNPs within species (*Massouras et al., 2012*); and (iii) within species, the number of 5′ variants is associated with ASE and multiple *cis*-regulatory variants occur within the same promoter region (*Renganaath et al., 2020*). One notable difference between ASE levels and dynamics is that 5′ InDels show a stronger association with ASE. Below, we discuss these results and some of the limitations of our study.

Based on the genome-wide logistic regression, we found InDels have larger odds ratios than SNPs, indicating either larger effects or a higher proportion of variants affect expression. This result is consistent with a larger effect of InDels found in an eQTL mapping study in *Drosophila* (*Massouras et al., 2012*). However, a limitation of our genome-wide associations is comparing 5′, coding and 3′ regions. First, the number of SNPs and InDels is correlated among 5′, coding and 3′ regions, making it hard to quantify their relative contribution. Second, the low odds ratios for 5′

compared to 3' region SNPs could be a consequence of the size of the region. The small (80 bp) 3' regions may have a higher proportion of functional variants than the larger 5' regions, which likely contain a mixture of large effect promoter variants diluted by more numerous non-functional variants outside of the promoter region. In a yeast eQTL study where direct comparison of 5' and 3' regions was possible, the strongest associations were for regions immediately upstream and downstream of the transcription start and termination sites, respectively (*Kita et al., 2017*). For the association with 3' variants, it is worth noting that ASE may be driven by variants that affect mRNA decay rather than transcription (*Cheng et al., 2017*).

We also found that 5' InDels have stronger associations with ASE dynamics compared to ASE levels. However, this difference was not present for genes with large allele differences, where both ASE levels and dynamics were strongly associated with 5' InDels (*Figure 3—figure supplement 1*). The strong and more equivalent associations could be a consequence of the higher coincidence of both ASE levels and dynamics for genes with large allele differences. We did not find an over-representation of InDels among causal variants identified in the CRE-seq assay. This difference may be a consequence of a small sample size (n = 35). But it could also reflect our selection of genes with the largest differences in ASE dynamics to test, and promoters with multiple SNPs being more likely to cause large ASE differences than those with multiple InDels, which are more rare.

A previous study of *cis*-regulatory variants that affect expression levels in yeast found multiple *cis*-regulatory variants per gene, and that *cis*-regulatory variants are more likely to disrupt conserved sequences and alter transcription factor binding sites (*Renganaath et al., 2020*). We found that nearly half of the genes (8/20) had more than one variant (2–5) associated with ASE dynamics, but *cis*-regulatory variants associated with ASE dynamics were not associated with conserved sequences or changes in predicted transcription factor binding sites. Beyond technical differences, the absence of association with conserved sequences and binding sites could be related to differences in *cis*-regulatory variants underlying ASE levels versus dynamics, to the strains used in each study, or to our smaller sample size. Strain differences may be relevant since we used variants between two wild strains Oak and ChII, whereas *Renganaath et al., 2020* used a wine and laboratory strain, the latter of which has evolved under relaxed selection and has more deleterious variants (*Gu et al., 2005*; *Doniger et al., 2008*). Consistent with a sample size explanation, we found that PhastCons conservation scores improved the odds ratios from genome-wide logistic regression of SNPs with ASE levels and dynamics (Table S6 in *Supplementary file 1*).

Although powerful in throughput, the CRE-seq reporter assay has a number of limitations relevant to our results. First, 130 bp CRE sequences do not capture the entire promoter and different regions of a promoter often generate different patterns of expression. For example, a variant that modulates expression may have little or no effect unless upstream activation sequences are also included in the CRE. However, it is also possible that a variant affects expression regardless of the presence or absence of other elements. The extent to which the effects of individual binding sites are dependent on other sites forms the basis for the difference between the enhanceosome and billboard models of *cis*-regulatory sequences (*Arnosti and Kulkarni, 2005*). A second limitation of our study is that high-throughput reporter assays perform better with high levels of replication. Prior high-throughput reporter assays have used tens or hundreds of barcode replicates per allele (*Tewhey et al., 2016*; *Renganaath et al., 2020*). Our use of only four barcode replicates per allele likely limited our ability to detect variants that affect ASE levels. This limitation applies less to ASE dynamics that are unaffected by the mean expression of any single barcoded CRE. Given these limitations, not all *cis*-regulatory variants assayed by CRE-seq may have been detected.

## Evolution of *cis*-regulatory sequences

Chimeric *cis*-regulatory sequence from different species often shows loss of function and supports the binding site turnover model, whereby the chance gain of a redundant binding site enables loss of another site without adverse effects on expression (*Ludwig et al., 2000*; *Ludwig et al., 2005*; *Arnold et al., 2014*). We find that chimeric promoter regions from *S. cerevisiae* and *S. uvarum* often generate expression dynamics outside of the parental species' range. The chimeras thus provide evidence for constraints on gene expression dynamics. However, we also find that *cis*-regulatory variants within *S. cerevisiae* are not greatly enriched at conserved sites. These two observations are consistent with a neutral model of expression divergence (*Fay and Wittkopp, 2008*), whereby small changes in dynamics within species are neutral, but when neutral changes in different lineages are

brought together they yield expression patterns that lie outside of the parent range and are unlikely to be tolerated within a species. It is also possible that constraints on expression dynamics depend on expression levels or noise. Indeed, the fitness effects of noise depend on expression levels (*Duveau et al., 2018*). This emphasizes the importance of characterizing how fitness altering *cis*-regulatory variants affect all aspects of gene expression to understand the evolution of *cis*-regulatory sequences.

# Materials and methods

## Key resources table

| Reagent type (species) or resource | Designation | Source or reference | Identifiers | Additional information |
|---|---|---|---|---|
| Gene (*Saccharomyces cerevisiae*) | ALD5 | Saccharomyces Genome Database | SGD:S000000875 | |
| Gene (*Saccharomyces cerevisiae*) | GND2 | Saccharomyces Genome Database | SGD:S000003488 | |
| Gene (*Saccharomyces cerevisiae*) | PHO3 | Saccharomyces Genome Database | SGD:S000000296 | |
| Strain, strain background (*Saccharomyces cerevisiae*) | Oak; YJF153 | PMID:12702333 | | YPS163 background |
| Strain, strain background (*Saccharomyces cerevisiae*) | Wine; YJF1442 | PMID:16103919 | | UCD2120 background |
| Strain, strain background (*Saccharomyces cerevisiae*) | ChI; YJF1373 | PMID:22913817 | | HN6 background |
| Strain, strain background (*Saccharomyces cerevisiae*) | ChII; YJF1375 | PMID:22913817 | | SX6 background |
| Strain, strain background (*Saccharomyces paradoxus*) | YJF694 | PMID:19212322 | | N17 background |
| Strain, strain background (*Saccharomyces uvarum*) | YJF1450 | PMID:22384314 | | CBS7001 background |
| Recombinant DNA reagent | pIM202 | PMID:23921661 | | CRE cloning vector |
| Sequence-based reagent | S288c genome | PMID:22384314 | | |
| Sequence-based reagent | N17 genome | PMID:22384314 | | |
| Sequence-based reagent | CBS 7001 genome | PMID:22384314 | | |
| Sequence-based reagent | Oak genome | This paper | YJF153.fasta; YJF153.gff | https://doi.org/10.17605/OSF.IO/Y5748 |

*Continued on next page*

*Continued*

| Reagent type (species) or resource | Designation | Source or reference | Identifiers | Additional information |
|---|---|---|---|---|
| Sequence-based reagent | Wine genome | This paper | BC217.fasta; BC217.gff | https://doi.org/10.17605/OSF.IO/Y5748 |
| Sequence-based reagent | ChI genome | This paper | HN6.fasta; HN6.gff | https://doi.org/10.17605/OSF.IO/Y5748 |
| Sequence-based reagent | ChII genome | This paper | SX6.fasta; SX6.gff | https://doi.org/10.17605/OSF.IO/Y5748 |
| Sequence-based reagent | Intra-specific CRE-library | This paper | CRE_Libraries.YJF1455.csv | https://doi.org/10.17605/OSF.IO/Y5748 |
| Sequence-based reagent | Inter-specific CRE-library | This paper | CRE_Libraries.YJF1484.csv | https://doi.org/10.17605/OSF.IO/Y5748 |
| Commercial assay or kit | Dynabeads mRNA Direct kit | Invitrogen | Invitrogen:61011 | |
| Commercial assay or kit | YeaStar DNA kit | Zymo Research | Zymo:D2002 | |
| Commercial assay or kit | YeaStar RNA kit | Zymo Research | Zymo:R1002 | |
| Commercial assay or kit | Dynabeads mRNA Direct kit | Zymo Research | Zymo:D2002 | |
| Commercial assay or kit | Glucose (GO) Assay Kit | Sigma | Sigma:GAGO20 | |
| Software, algorithm | BWA v0.7.5 | PMID:19451168 | RRID:SCR_010910 | https://github.com/lh3/bwa |
| Software, algorithm | PicardTools v1.114 | Broad Institute | RRID:SCR_006525 | https://github.com/broadinstitute/picard |
| Software, algorithm | GATK Haplotype Caller v3.3–0 | Broad Institute | RRID:SCR_001876 | https://github.com/broadinstitute/gatk/ |
| Software, algorithm | liftOver | UCSC Genome Browser | RRID:SCR_018160 | https://genome-store.ucsc.edu/ |
| Software, algorithm | Fastx-toolkit | Hannon Lab | RRID:SCR_005534 | https://github.com/agordon/fastx_toolkit |
| Software, algorithm | Bowtie2 v2.1.0 | PMID:22388286 | RRID:SCR_016368 | https://github.com/BenLangmead/bowtie2 |
| Software, algorithm | Htseq-count | PMID:25260700 | RRID:SCR_011867 | https://github.com/htseq/htseq |
| Software, algorithm | DESeq2 | PMID:25516281 | RRID:SCR_015687 | https://doi.org/10.18129/B9.bioc.DESeq2 |
| Software, algorithm | Patser | PMID:10487864 | | http://stormo.wustl.edu/software.html |
| Software, algorithm | Custom R scripts | This paper | | https://doi.org/10.17605/OSF.IO/Y5748 |

## Strains

Three strains of *S. cerevisiae,* one of *S. paradoxus,* and one of *S. uvarum* were crossed to a common reference YJF153, a derivative of an *S. cerevisiae* oak isolate from North America (Table S1 in *Supplementary file 1*). The three intra-specific hybrids were generated through crosses to a wine

strain from North America (UCD2120 × YJF153 = YJF1460) and two wild strains from China (HN6 × YJF153 = YJF1454 and SX6 × YJF153 = YJF1455). Strains were chosen to reflect a range of divergence. The two strains from China are from two of the most divergent *S. cerevisiae* lineages: China I (HN6) and China II (SX6) (*Wang et al., 2012*). The inter-specific hybrids were generated using *S. paradoxus* (N17 × YJF153 = YJF1453) and *S. uvarum* (CBS7001 × YJF153 = YJF1484). Diploid hybrids were generated by mixing strains of opposite mating type and selecting for dominant drug resistance markers present at the *HO* locus.

## RNA-sequencing time course

Each hybrid strain was cultured in 125 mL YPD at 250 rpm at 30°C with an initial density of ~3 × 10$^6$ cells/mL. Approximately 3 × 10$^8$ cells were taken at each timepoint, centrifuged, supernatant removed, and flash frozen in liquid nitrogen. A total of 19 samples were collected during the switch from fermentation to respiration with the most intense sampling occurring every 15 min after glucose depletion (*Figure 1—figure supplement 1*). Glucose depletion was measured using a Glucose (GO) Assay Kit (Sigma-Aldrich). RNA was extracted with phenol-chloroform and mRNA purified by oligo-dT (Dynabeads mRNA Direct kit, Invitrogen). cDNA libraries were made by reverse transcription, fragmentation, and adaptor ligation by Washington University's Genome Technology Access Center (GTAC). Adaptors contained 7 bp indexes for multiplexing samples. The pooled equimolar libraries of 95 samples (19 sampling time × 5 strains) were paired-end sequenced (2 × 40 bp) using three runs of an Illumina NextSeq. A total of 1255.9 million paired-end reads were generated.

## Reference genomes and variants

The genomes of the three *S. cerevisiae* strains were sequenced to generate reference genomes for mapping RNA-sequencing reads and to identify variants associated with ASE. DNA was extracted (YeaStar DNA kit, Zymo Research), libraries were generated by GTAC, and paired-end (2 × 101 bp) reads were generated using an Illumina HiSeq 2500, resulting in 5.2–5.9 million paired reads per strain. Reads were mapped to the S288c reference genome (R64-1-1) using BWA v0.7.5 (*Li, 2013*), and duplicates were marked using PicardTools v1.114 (http://broadinstitute.github.io/picard/). SNPs and InDels were called using GATK's HaplotypeCaller v3.3-0 (*Van der Auwera et al., 2013*) following InDel realignment, base recalibration, and variant recalibration using an independently derived set of SNPs and InDels. The independent set of SNPs (21,327) and InDels (4748) was identified using GATK after BWA mapping of 27 assembled genomes (Table S7 in *Supplementary file 1*) to the S288c reference genome. Variants were filtered to remove variable sites with calls in fewer than 20 strains and minor allele frequencies of less than 15%. Using GATK's tranche filter set to 99.9 (percent sensitivity to the independent set of variants), we identified 222,589 SNPs and 20,485 InDels within the three *S. cerevisiae* strains and S288c. The Oak strain is most closely related to the Wine strain (76,659 variants), followed by China II (111,591 variants) and China I (128,346 variants).

Variants were used to generate reference genomes for mapping RNA-sequencing reads. Reference genomes were generated using the S288c genome as a template and GATK's FastaAlternateReferenceMaker command to incorporate variants present in each strain. Genome annotations were generated using liftOver (*Hinrichs et al., 2006*) to transfer S288c annotations to each of the three other genomes. For mapping inter-specific hybrid reads, we used reference genomes and annotations (*Scannell et al., 2011*) for *S. cerevisiae* (S288c), *S. paradoxus* (N17), and *S. uvarum* (CBS7001).

## Gene expression measurements

RNA-sequencing reads were mapped to combined reference genomes containing the genomes of both parental strains used to generate the hybrid in order to avoid mapping bias. Reads were demultiplexed using the Fastx-toolkit (http://hannonlab.cshl.edu/fastx_toolkit/) and then mapped to combined reference genomes using Bowtie2 v2.1.0 (*Langmead and Salzberg, 2012*) using the local alignment setting (–local) and a maximum of one mismatch in the seed alignment (-N 1). Duplicate reads were marked using PicardTools. Htseq-count (*Anders et al., 2015*) was used to quantify ASE by counting reads that mapped to each allele in the combined reference genomes. Reads mapping to overlapping features (-mode union) and reads with a mapping quality less than 10 (default) were not counted. Thus, reads that mapped equivalently to both alleles of a gene and assigned mapping quality of 0 or 1 by Bowtie2 were removed. For inter-specific hybrids, we used previous definitions

of orthologous genes (*Scannell et al., 2011*). The fraction of mappable reads used for ASE was 81.58%, 82.62%, and 83.11% for the intra-specific hybrids YJF1454, YJF1455, and YJF1460, and 86.16% and 86.57% for the inter-specific hybrids YJF1450 and YJF1484, respectively. None of the intra-specific hybrids showed any mapping bias, except in YJF1454 genes on chromosome XIII R showed uniformly higher expression of the Oak compared to the ChI allele consistent with aneuploidy (*Figure 1—figure supplement 2*). The fraction of reads mapping to the common reference YJF153 was 52.58% (50.59%, after removing chrXIII R), 50.11%, 49.78%, 49.11%, and 46.90% for YJF1454, YJF1455, YJF1460, YJF1453, and YJF1484. The median number of allele-specific reads per timepoint was 2.9 million reads (range: 1.5–4.8) for intra-specific hybrids and 2.0 million reads (range: 1.4–3.3) for inter-specific hybrids.

## Statistical analysis of differentially expressed alleles

ASE counts were normalized using DESeq's blind method (*Love et al., 2014*). Differences in expression levels were tested using a weighted linear model: $f_i = 0.5 + \beta_0 + e$, where $f_i$ is the normalized frequency of the YJF153 reference allele over timepoint $i$, $\beta_0$ is the deviation from 0.5, and $e$ is the error with weights based on the total number of reads at each timepoint. Differences in expression dynamics were tested using a weighted Durbin–Watson test for an autocorrelation across timepoints of the allele differences. The weighted Durbin–Watson test is based on the *lmtest* package of R where the residuals ($e$) from the same linear model used above are used to calculate the test statistic:

$$weighted - dw = \frac{\sum_{i=1}^{T-1}(w_{i+1}e_{i+1} - w_i e_i)^2}{\sum_{i=1}^{T}(w_i e_i)^2}$$

where $w_i$ and $e_i$ are the weights (read counts) and residuals for sample $i$ out of $T$ timepoints. For both tests, we used FDR cutoffs of 0.01. For all analyses, FDR was calculated using the method of *Benjamini and Hochberg, 1995*.

Ten permutations of the data were used to validate the statistical cutoffs. Permuting the counts for the two alleles independently at each timepoint yielded an average of between 0.3 and 2.0 false positive across the five hybrids at an FDR cutoff of 0.01 for the test of ASE levels. Permuting the timepoints yielded an average of between 2.3 and 7.7 false positives across the five hybrids at an FDR cutoff of 0.01 for the test of ASE dynamics.

Genes with low read counts were removed from the analysis: either those with less than an average of 20 reads per timepoint or more than seven timepoints with no reads. This filter eliminated an average of 692 genes per hybrid and left a total of 4703 genes with data in all five hybrids.

## Gene expression clusters

k-means clustering was used to group genes by their combined allele expression profile and by their allele imbalance profile. For the combined gene expression profiles, 19,633 genes from all five hybrids with a significant (FDR < 0.01) autocorrelation in the combined (both alleles) profile over time were normalized (centered and scaled) and clustered into 12 groups. For the allele imbalance profiles, allele frequencies were centered but not scaled for genes with significant ASE dynamics (n = 6135) and clustered into 12 groups. For both, 336 genes without any reads at one or more timepoints were removed since k-means clustering does not handle missing data.

## Association with variants

The numbers of SNPs and InDels for intra-specific hybrids were obtained from variant calling, and the corresponding counts for inter-specific hybrids were obtained from multiple sequence alignments without correction for multiple hits (*Scannell et al., 2011*). Upstream intergenic regions were defined by sequences between adjacent coding regions, except for cases of short (< 5 bp) or overlapping coding sequences where we extended the region to the next upstream gene. Downstream intergenic regions were defined as 80 bp downstream of the stop codon. Coding regions included all variants within them. Variant counts were obtained for all of the 4703 genes with expression data except for 11 genes that only had counts for the intra-specific hybrids.

For logistic regression, we predicted genes with differences in ASE levels and dynamics using $logit(p) = b_0 + b_1 x + e$, where $p$ indicates whether a gene shows significant ASE or not, $b_1$ is

the regression coefficient, $x$ is the predictive variable, and $e$ is the error. We used a Bonferroni cutoff for significance to correct for the 60 different regressions based on 6 predictive variables: number of SNPs and InDels within upstream, downstream, and coding regions, and 10 response variables: ASE dynamics and ASE levels in each of the five hybrids.

## CRE-seq

A high-throughput reporter assay (CRE-seq) was used to measure the activity of CREs by sequencing (*Mogno et al., 2013*). In this assay, a pool of synthetic *cis*-regulatory sequences are cloned en masse, YFP is inserted between the CRE and the barcode, and the reporter library is then integrated into the genome. The activity of each CRE is measured by the ratio of barcode sequencing reads from an RNA relative to a DNA library of the pooled transformants. As described below, we used this reporter assay to measure expression of 7268 and 7232 synthetic CRE sequences, representing intra-specific and inter-specific allele differences upstream of 69 and 98 genes, respectively.

## Synthesis

CREs were synthesized as part of a library of 200 bp oligos (Agilent). The synthetic oligos include a forward primer, RS1, CRE, RS2, RS3, RS4, BC, RS5, reverse primer, where CRE is the *cis*-regulatory element, RS1–5 are restriction sites, and BC is a barcode (*Figure 4—figure supplement 1*). The barcodes were random 10 bp sequences with a minimum of two differences and no restriction sites. The selected barcodes were checked to ensure no base composition bias. Target CRE sequences were defined by the 250 bp region upstream of the TSS (*Venters et al., 2011*). For each target CRE, five 130 bp sliding windows of the sequence were generated using a 30 bp step size. Any 130 bp regions containing coding sequence of the next upstream gene were removed. Four replicates with different barcodes were generated for each CRE. CREs were generated separately for each strain allele unless the CRE window was identical between the strains. TSSs (*Xu et al., 2009*) were found by liftOver from the reference (S288c) genome to the coordinates in the strain of interest. In cases where the CREs contained InDels, extra sequence was added to the gapped allele between RS2 and RS4 to keep the oligo length constant. CREs were also synthesized for intra-specific variants between the Oak (YJF153) and China II strain (SX6). These CREs were designed by replacing a single Oak variant by the China II variant and vice versa (*Figure 4—figure supplement 1*). Inter-specific chimeric CREs were generated by recombining the *S. cerevisiae* allele with *S. uvarum* allele at the center of each CRE (*Figure 4—figure supplement 1*). The intra-specific and inter-specific libraries respectively represented 334 and 452 regions upstream of 69 and 98 genes after removing regions that overlapped with upstream genes, and contained a total of 7268 and 7232 synthetic CRE sequences. Genes were chosen based on:the absence of restriction sites within the target CRE, an annotated TSS, significant ASE levels and/or ASE dynamics, and inspection of the ASE differences. For the 69 genes used for the intra-specific libraries, 19 showed ASE levels, 10 showed ASE dynamics, and 40 showed both. For the 98 genes used in the inter-specific library, 23 showed ASE level, 10 showed ASE dynamics, and 65 showed both.

## Cloning

CRE-seq libraries were amplified (eight separate reactions of 10-cycle amplification), digested, gel purified, and ligated into pIM202 (*Mogno et al., 2013*). This differs from the original protocol that used higher-resolution acrylamide gel extraction since sequencing the amplified library showed only a small portion (<1%) of sequences were outside of the 200+/−5 bp resolution of the acrylamide gel. Ligated products were transformed into bacteria using electroporation and 80–100k colonies were pooled and the plasmid library was extracted. YFP along with 69 bp of the *TSA1* core promoter (*Mogno et al., 2013*) was inserted between the CRE and the barcode and then transformed back into bacteria by electroporation. At each step, PCR was used to ensure a low frequency of empty vector.

## Transformation into yeast

The reporter gene (YFP) with synthetic CREs and corresponding barcodes were integrated at the *URA3* locus (*Figure 4—figure supplement 1*). BstBI-digested plasmids were transformed into the Oak strain (YJF186) using the LiAc method (*Gietz et al., 1995*). We collected ~100k colonies on

complete minimal plates without uracil, and integration was confirmed by PCR for 20/20 randomly selected colonies. The yeast strains transformed with intra-specific library were co-cultured with SX6 (YJF1375) in YPD at 30°C overnight to allow mating, and HygB and dsdA were used to select diploid cells. The pool of yeast strains transformed with inter-specific library was co-cultured with the *S. uvarum* strain (YJF1450) in YPD overnight at 30°C to allow mating. Double selection with nourseothricin and dsdA was used to select diploid cells.

## Full-length promoters

Three genes, *ALD5*, *GND2,* and *PHO3*, were chosen to compare short- and full-length promoter constructs. Full-length promoters from the Oak (YJF186) and ChII (SX6) strains were amplified from the TSS to the next upstream coding sequence. Four barcodes per construct and restriction sites were incorporated into the primers used for PCR. Each full-length promoter with barcodes was cloned into the pIM202 plasmid and YFP was inserted between the full-length promoter and the barcode. BstBI-digested plasmids were transformed into the Oak strain at the *URA3* locus. Each of the transformed strains were mated with SX6 to form diploids.

## CRE-seq expression measurements

Pooled libraries were shaken at 250 rpm in 125 mL YPD at 30°C with an initial density of $3 \times 10^6$ cells/mL. Samples of $3 \times 10^8$ cells were taken for DNA and RNA extraction between 6 hr and 15 hr, which spans the switch from fermentation to respiration. A total of 19 samples were taken for RNA measurements for the intra-specific library, and a total of 27 samples were taken for the inter-specific library. For both libraries, sampling corresponded to the RNA-sequencing timepoints but with more dense sampling for the inter-specific library. For each library, samples were taken for DNA measurements at the first and last timepoints. Cells were centrifuged for 30 s at 1000 g, and the pellets were immediately frozen with liquid nitrogen.

The abundance of each CRE in the library was measured by sequencing the barcodes from DNA, and the expression of each CRE was measured by sequencing the barcodes from RNA extracts. DNA was extracted from the first and the last timepoints using YeaStar Genomic DNA Kit (Zymo Research). RNA was extracted by YeaStar RNA Kit (Zymo Research) and residual DNA digested with DNase (Promega). mRNA was purified by oligo-dT (Invitrogen), and cDNA was made using Super-Script II Reverse Transcriptase (Invitrogen). mRNA was removed after first-strand DNA synthesized with RNase H (New England BioLabs). Combinations of indexed Ion Torrent primers were used to amplify barcodes in the library for each timepoint (*Figure 4—figure supplement 1*) using Phusion High-Fidelity PCR (New England BioLabs). To avoid sampling bias during PCR amplification, each sample was amplified with 4 PCR reactions and 20 cycles and then pooled. PCR pools were gel-extracted and cleaned individually, then pooled together at equivalent concentrations. Libraries were sequenced on an Ion Torrent Proton machine at St. Louis University's Genomics Core Facility.

After demultiplexing the samples using the indexed Ion Torrent primer pairs, perfect matches to the CRE barcodes were found for 401 and 97 million reads from the intra- and inter-specific libraries. For one sample, three technical replicate libraries was generated (PCR and sequencing). The technical replicates showed an average correlation of 0.96 for barcode counts and were subsequently combined. We removed CREs that had either zero reads in more than a third of the RNA timepoint samples or those with less than 100 reads on average in either the RNA or DNA samples. This filter left 6820 intra-specific and 5013 intra-specific CREs with an average of 16.4 and 3.2 million reads per timepoint and a median of 1278 and 357 reads per barcode across all timepoints for the intra- and inter-specific libraries, respectively. The read count distribution of intra- and inter-specific libraries was normalized using DESeq2's blind method (*Love et al., 2014*). Expression of each CRE allele was measured by the ratio of RNA to DNA counts. Barcodes with zero counts were treated as missing data in statistical models.

## Identification of significant expression differences

CRE-seq regions were tested for correlations (Pearson) with RNA-seq using the average expression of all barcodes for a given CRE region. For inter-specific CRE-seq, correlations were measured between the 19 timepoints closest to the RNA-seq timepoints. CREs with significant differences in expression levels and dynamics were tested using the weighted linear model and weighted Durbin–

Watson test, respectively. When testing for differences in expression levels using the linear model, we used the average expression across all timepoints for each barcode rather than treating each timepoint as an independent measure of expression levels. We chose this more conservative test to avoid cases where one barcoded CRE had substantially higher or lower expression across all time-points compared to the other replicate barcodes. Consequently, the power to detect differences in average CRE expression levels based on four barcode replicates was reduced. The test for expression dynamics was not affected by this problem since expression dynamics was measured by changes in the ratio of expression levels over time.

## Identification of variant and chimera differences

For the intra-specific library, we tested individual variants for those CREs with more than one difference between the Oak and ChII alleles. We used the same weighted tests for differences in expression levels and dynamics, but tested each variant genotype separately (*Figure 6—figure supplement 1*). For the inter-specific library, we used the chimeric CREs to map differences to the proximal or distal part of each region showing significant differences between the parental *S. cerevisiae* and *S. uvarum* alleles. The left (distal) and right (proximal) promoter regions were separately tested using the genotype of the left and right portions of the CRE, respectively (*Figure 6—figure supplement 1*). Parental differences were classified as mapping to the proximal region if the proximal but not the distal genotype was significant. Distal mapping was similarly classified. Chimeric CREs that showed significant differences from both parents were also identified, and subsequently classified as outside or inside the parental range if their average expression distance (Euclidean) to each parent was greater or less than the distance between the two parents, respectively. A flow diagram of the number of CREs tested along with genotypes tests for significant variants and chimeras is presented in *Figure 6—figure supplement 1*.

## Associations with variant annotations

Variants were annotated with PhastCons and transcription factor binding site scores. PhastCons scores (*Siepel et al., 2005*) were obtained for yeast from the USCS Genome Browser. Scores were extracted for each SNP, and the average score was obtained for InDels based on the two sites flanking the InDel and any scores within the InDel. Transcription factor binding motifs were obtained for 196 factors (*Spivak and Stormo, 2012*). For each variant, we extracted 30 bp of sequence from the Oak and ChII genome on either side of the variant. We used Patser (*Hertz and Stormo, 1999*) to scan each sequence with each motif model and the best hit was recorded. A background nucleotide frequency of 36% GC was used, and scores less than zero (equivalent likelihood between the motif and background model) were set to zero. The difference in score between the two alleles was calculated for each motif model, and the maximum difference across all motif models was used as the binding site annotation score for each variant. PhastCons scores and binding site scores were each tested for association with CRE variants that were positive (n = 35), negative (n = 35), and all other intergenic variants (n = 44,514) by ANOVA.

## Acknowledgements

We thank Feng-Yan Bai for sharing yeast strains and members of the Fay lab for their suggestions and comments.

## Additional information

### Funding

| Funder | Grant reference number | Author |
| --- | --- | --- |
| National Institutes of Health | GM080669 | Justin Fay |

The funders had no role in study design, data collection and interpretation, or the decision to submit the work for publication.

## Author contributions
Ching-Hua Shih, Data curation, Software, Formal analysis, Investigation, Writing - original draft, Writing - review and editing; Justin Fay, Conceptualization, Formal analysis, Funding acquisition, Writing - original draft, Writing - review and editing

## Author ORCIDs
Justin Fay https://orcid.org/0000-0003-1893-877X

## Decision letter and Author response
Decision letter https://doi.org/10.7554/eLife.68469.sa1
Author response https://doi.org/10.7554/eLife.68469.sa2

# Additional files

## Supplementary files
• Supplementary file 1. Supplementary tables. Table S1: strains used in this study. Table S2: k-means clusters of gene expression dynamics. Table S3: k-means clusters of allelic differences in expression. Table S4: logistic regression of allele-specific expression (ASE) dynamics and levels. Table S5: average number of single-nucleotide polymorphism (SNP) and insertion/deletion (InDel) differences within hybrids. Table S6: logistic regression with binding site and conservation scores. Table S7: genome assemblies used to identify variants.

• Transparent reporting form

## Data availability
Genome sequencing and assembly data were deposited into NCBI, see Table S1 and S7 in Supplementary file 1 for accessions. RNA sequencing data were deposited into NCBI's GEO database under GSE165594. Analysis scripts, data and summary files are available at https://doi.org/10.17605/OSF.IO/Y5748.

The following datasets were generated:

| Author(s) | Year | Dataset title | Dataset URL | Database and Identifier |
|---|---|---|---|---|
| Shih CH, Fay J | 2021 | Cis-regulatory variation affects gene expression dynamics | https://www.ncbi.nlm.nih.gov/geo/query/acc.cgi?acc=GSE165594 | NCBI Gene Expression Omnibus, GSE165594 |
| Shih CH, Fay J | 2021 | Gene expression dynamics | https://osf.io/y5748/ | Open Science Framework, 10.17605/OSF.IO/Y5748 |

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
