## [Decision Letter]

**Acceptance summary:**

The authors use RNAseq in yeast hybrids to study the effect of cis-variation on evolutionary divergence in gene expression and expression dynamics. What sets this study apart from previous work is that the authors use hybrids across different genetic distances, separate expression levels and dynamics by sampling across different time points during an environmental shift, and also investigate 3' sequences. The main conclusions confirm that SNPs and InDels both affect gene expression as well as dynamics, and that on average, InDels have larger effects compared to SNPs, especially on expression dynamics. Moreover, the results also reveal negative selection on expression levels, with the effect of some cis mutations compensated by other cis variation, which ultimately results in complex interactions between the different cis-acting polymorphisms. Together, the results further our understanding of how cis sequence variation supports divergence in gene expression levels and dynamics.

**Decision letter after peer review:**

Thank you for submitting your article *Cis*-regulatory variants affect gene expression dynamics in yeast" for consideration by *eLife*. Your article has been reviewed by 3 peer reviewers, one of whom is a member of our Board of Reviewing Editors, and the evaluation has been overseen by Naama Barkai as the Senior Editor. The reviewers have opted to remain anonymous.

In general, this is a solid and interesting study. None of the reviewers have suggested additional experiments, but each reviewer did suggest a few tweaks to the data analysis and discussion.

1. Please consider to also take the effect size into account (i.e. by how much a certain genetic variation affects transcription). This could be done in several ways, the easiest one perhaps by repeating some analysis only for alleles that have a strong influence and comparing the results to the complete dataset.

2. Reconsider the statistics as suggested by reviewer 3.

3. Please expand a few sections to improve clarity, as suggested by the different reviewers, including a better discussion of the results in the light of what was already known from previous studies (eg work by Naama Barkai's team) or generally expected (namely that cis variation affects gene expression).

The individual reviews are found below. They give more details and a few comments that, although not essential, you might also want to consider when revising your manuscript.

*Reviewer #1 (Recommendations for the authors):*

It would be useful to give the reader an idea of the average magnitude of divergence in expression levels and dynamics. What is the average difference in expression level (average % difference across all time points) or dynamics (maximal % difference at any given time). I wonder whether it is then possible to filter out mutations that likely have a very large effect on transcription levels or dynamics? As far as I can tell, all variations are lumped together, irrespective of the magnitude of their effect. What happens if you study the effect of variations that occur around genes that show the largest divergence in expression levels or dynamics? I understand that in the interspecific hybrids, there may be too many variants linked to one gene, but in the intraspecific hybrids, it may be possible to identify variants that are linked to genes that show very large expression divergence

I am not sure whether I get Figure 2. Is it really true that the odds ratio is largest for 3' sequences? If so, that merits more discussion I think, as most readers would likely expect that 5' differences (in the promoter rather than the terminator) would have larger differences.

At first sight, the novel insight generated by this study might be quite limited? It would be useful to explicitly discuss which of their results confirm previous findings, and which are novel (such as perhaps the effect size of 5' vs 3' variation?)

*Reviewer #3 (Recommendations for the authors):*

I list my comments in rough order of importance:

– I missed an analysis of the types of genes that were subject to dynamic ASE. Were these genes enriched in certain functional categories? Were they more or less likely to be essential? More or less conserved at the protein level?

– Line 380: This section was not clear to me. What is the purpose of the "known" variants? That they determine thresholds for declaring new variants as true positives? The description of the "tranche filter" in line 386 did not clarify this. I suggest expanding or clarifying this section.

– Line 428: Provide information on how FDR was calculated.

– Line 535: Unless I completely missed it, I did not see information on how the CRE-Seq libraries were sequenced? Technology, machine, read length,…?

– Line 134: 15 SNPs and 107 Indels must be the wrong way around. Probably there are more SNPs than indels? That would also better match the numbers presented in the Methods.

– Abstract, line 13: The "but" connecting the two clauses seems off. The two statements aren't really in contrast, they're just two different statements.

– Line 215: "gaps" sounds like a reference genome assembly gaps, why not call this "indels"?

– In Table 1, I'd put the "Both" row at the bottom, not in between the two rows it combines.

---

## [Author Response]

Reviewer #1 (Recommendations for the authors):It would be useful to give the reader an idea of the average magnitude of divergence in expression levels and dynamics. What is the average difference in expression level (average % difference across all time points) or dynamics (maximal % difference at any given time). I wonder whether it is then possible to filter out mutations that likely have a very large effect on transcription levels or dynamics? As far as I can tell, all variations are lumped together, irrespective of the magnitude of their effect. What happens if you study the effect of variations that occur around genes that show the largest divergence in expression levels or dynamics? I understand that in the interspecific hybrids, there may be too many variants linked to one gene, but in the intraspecific hybrids, it may be possible to identify variants that are linked to genes that show very large expression divergence.

Figure S3 is intended to provide some indication of the effect size for expression levels (average ASE deviation from 0.5) and expression dynamics (ASE standard deviation over time). We used the standard deviation rather than the maximum to avoid outlier time-points. From this we see that most differences are small, <5%, but there is a long tail of larger effects.

To examine how the magnitude of ASE influences the odds ratios we split the data into three equal groups of genes based on the sum of squared deviations from a 0.5 allele frequency. We used this criteria rather than two separate measures for ASE levels and dynamics since the latter would have resulted in comparison of odds ratios from a different set of genes for ASE levels and dynamics. The results from genes with small, medium and large ASE are similar except that InDels are better predictors of dynamics compared to levels for genes with small ASE differences, but InDels have similar predictive ability for genes with large ASE differences. We have now added the analysis of effect size along with a figure (Figure 2—figure supplement 1). We also point out in the discussion that genes with large ASE difference are more likely to exhibit both ASE levels and ASE dynamics, thereby explaining the similar associations.

Within species there are some large effect genes with only a single upstream variant. However, because we couldn't exclude coding or 3' variants we didn't do any further analysis of these potentially large effect variants.

In examining the odds ratios for different effect sizes, we noticed subtle differences between the values in Figure 2 and the supporting tables. The original supporting tables have the correct numbers and we have updated Figure 2 accordingly. These differences don't affect our interpretations or conclusions.

I am not sure whether I get Figure 2. Is it really true that the odds ratio is largest for 3' sequences? If so, that merits more discussion I think, as most readers would likely expect that 5' differences (in the promoter rather than the terminator) would have larger differences.

Correct, but comparing regions is problematic. The 3' region is small (80 bp) and may have a higher proportion of functional variants compared to the 5' region, which include all variants between the two genes. Thus, comparison between regions is problematic. We have removed such comparisons from the results and have added this point to the discussion: "the low odds ratios for 5' compared to 3' region SNPs could be a consequence of the size of the region. The small (80 bp) 3' regions may have a higher proportion of functional variants than the larger 5' regions, which likely contain a mixture of large effect promoter variants diluted by more numerous non-functional variants outside of the promoter region."

At first sight, the novel insight generated by this study might be quite limited? It would be useful to explicitly discuss which of their results confirm previous findings, and which are novel (such as perhaps the effect size of 5' vs 3' variation?)

Our work is novel in that we examine expression dynamics, not just expression levels. Our results show these two aspects of expression are largely similar to one another and so our insight into *cis*-regulatory variation is somewhat limited. Nevertheless, we believe the results to be important for future studies of *cis*-regulatory variants and how they affect gene expression.

We have added explicit statements in the discussion regarding results that have been found before. "Previous studies have characterized cis-regulatory variants underlying gene expression levels. […] One notable difference between ASE levels and dynamics is that 5' InDels show a stronger association with ASE. Below, we discuss these results and some of the limitations of our study."

Reviewer #3 (Recommendations for the authors):I list my comments in rough order of importance:– I missed an analysis of the types of genes that were subject to dynamic ASE. Were these genes enriched in certain functional categories? Were they more or less likely to be essential? More or less conserved at the protein level?

Genes with ASE dynamics were enriched for ribosome biogenesis, carboxylic acid metabolic process and oxidation-reduction process, as might be expected for genes that change during the diauxic shift. We did not include a characterization of genes (GO, essential, conservation) as this was not an objective of our study. However, the list of significant genes for each hybrid is available on the OSF repository (FigureS3_data.csv).

– Line 380: This section was not clear to me. What is the purpose of the "known" variants? That they determine thresholds for declaring new variants as true positives? The description of the "tranche filter" in line 386 did not clarify this. I suggest expanding or clarifying this section.

We clarified that an independent set of variants was used to determine thresholds for calling new variants as true positives based on 99.9 sensitivity. "Single nucleotide polymorphisms (SNPs) and insertions/deletions (InDels) were called using GATK’s HaplotypeCaller v3.3-0 (Van der Auwera et al., 2013) following InDel realignment, base recalibration and variant recalibration using an independently derived set of SNPs and InDels. […] Using GATK's tranche filter set to 99.9 (percent sensitivity to the independent set of variants), we identified 222,589 SNPs and 20,485 InDels within the three *S. cerevisiae* strains and S288c."

– Line 428: Provide information on how FDR was calculated.

We now state: "FDR was calculated using the method of Benjamini and Hochberg (1995)."

– Line 535: Unless I completely missed it, I did not see information on how the CRE-Seq libraries were sequenced? Technology, machine, read length,…?

We now state: "Libraries were sequenced on an Ion Torrent Proton machine at St. Louis University's Genomics Core Facility." We did not included read length because length varies within a run on the Proton platform.

– Line 134: 15 SNPs and 107 Indels must be the wrong way around. Probably there are more SNPs than indels? That would also better match the numbers presented in the Methods.

Corrected.

– Abstract, line 13: The "but" connecting the two clauses seems off. The two statements aren't really in contrast, they're just two different statements.

Corrected.

– Line 215: "gaps" sounds like a reference genome assembly gaps, why not call this "indels"?

Corrected.

– In Table 1, I'd put the "Both" row at the bottom, not in between the two rows it combines.

Done.